# Impact of Vitamin B12 Insufficiency on the Incidence of Sarcopenia in Korean Community-Dwelling Older Adults: A Two-Year Longitudinal Study

**DOI:** 10.3390/nu15040936

**Published:** 2023-02-13

**Authors:** Seongmin Choi, Jinmann Chon, Seung Ah Lee, Myung Chul Yoo, Sung Joon Chung, Ga Yang Shim, Yunsoo Soh, Chang Won Won

**Affiliations:** 1Department of Physical Medicine and Rehabilitation Medicine, Kyung Hee University Hospital, Seoul 02447, Republic of Korea; 2Department of Physical Medicine and Rehabilitation, Kyung Hee University Hospital at Gangdong, Seoul 05278, Republic of Korea; 3Department of Family Medicine, College of Medicine, Kyung Hee University, Seoul 02447, Republic of Korea

**Keywords:** vitamin B12, cobalamin, sarcopenia, the short physical performance battery

## Abstract

The longitudinal effect of B12 insufficiency on sarcopenia has not yet been investigated in older adults. We aimed to study the impact of B12 levels on alterations in muscle mass, function and strength over two years. Non-sarcopenic older adults (*n* = 926) aged 70–84 were included. Using the Korean Frailty and Aging Cohort Study, this two-year longitudinal study used data across South Korea. The tools used for assessing muscle criteria were based on the Asian Working Group for Sarcopenia guidelines. Participants were divided into the insufficiency (initial serum B12 concentration < 350 pg/mL) and sufficiency groups (≥350 pg/mL). Logistic regression analyses were performed to evaluate the effect of initial B12 concentration on sarcopenia parameters over a two-year period. In women, multivariate analysis showed that the B12 insufficiency group had a significantly higher incidence of low SPPB scores (odds ratio [OR] = 3.28, 95% confidence interval [CI] = 1.59–6.76) and sarcopenia (OR = 3.72, 95% CI = 1.10–12.62). However, the B12 insufficiency group did not have a greater incidence of sarcopenia or other parameters in men. Our findings suggest B12 insufficiency negatively impacts physical performance and increases the incidence of sarcopenia only in women.

## 1. Introduction

Vitamin B12 (B12), or cobalamin, is a water-soluble vitamin B complex member. It is a coenzyme involved in DNA synthesis and in fat and amino acid metabolism in the body [1]. B12, referred to as a “neurotropic” vitamin, plays an essential role in both the peripheral nervous system and the central nervous system, including the brain and spinal cord, so a supply of B12 is fundamental for the normal function of the nervous system [2,3]. A deficiency of B12 promotes neurotoxic oxidative stress and even neurodegeneration, which could be a risk factor for cognitive impairment [4]. B12 deficiency can also lead to the development of disorders of the peripheral nerves, such as peripheral neuropathy [5]. B12 deficiency is common among older adults because of the high prevalence of malabsorption due to atrophic gastritis and long-term antacid therapy, including histamine H_2_ blockers or proton pump inhibitors. In addition, with aging, gastric acid secretion and pepsin levels decrease and the absorption of B12 is reduced in older patients [6,7]. Demyelinating neurologic disorders in the central and peripheral nervous system stemming from B12 deficiency in older patients and significant damage of nerve fibers could cause muscle weakness, numbness, imbalance, ataxia and even reduced skeletal muscle mass [8].

Loss of skeletal muscle mass is manifested as sarcopenia in older adults, characterized by decreased muscle strength and/or decreased physical function causing disability and even increased mortality [9]. According to a recent study in Korea, the prevalence of sarcopenia is approximately 21.3% in men and 13.8% in women aged 70–80 years in Korea [10]. The underlying causes of sarcopenia include nutritional (low protein intake, energy intake, micronutrient deficiency), inactivity (bed rest, low activity) and diseases (endocrine diseases, neurological disorders, cancer) [11]. Because B12 deficiency contributes to neurodegeneration, a previous study suggested that this defect may be related to sarcopenia in older adults [8]. In particular, cognitive dysfunction and peripheral neuropathy related to B12 deficiency are known to be risk factors for sarcopenia [12,13]. Our previous cross-sectional study had found that B12 deficiency may increase the prevalence of low skeletal muscle mass in older adults [14]. However, these studies were cross-sectional in nature and therefore it was difficult to investigate the longitudinal association between B12 deficiency and sarcopenia.

In this two-year longitudinal study, we aimed to examine the impact of low B12 levels on the incidence of sarcopenia in healthy community-dwelling older adults using dual-energy X-ray absorptiometry (DEXA). We used baseline data from the Korean Frailty and Aging Cohort Study (KFACS) in this longitudinal study.

## 2. Materials and Methods

### 2.1. Data and Study Population

This two-year longitudinal study used data from the 2016 to 2019 KFACS. The KFACS is a prospective cohort, a nationwide study conducted in eight medical and two public health centers across South Korea. The authors recruited community-dwelling older adults between 70 and 84 years of age for two years (2016–2017) and followed up with them after two years (2018–2019) with 4 months of allowance limitations. In the baseline survey, questionnaires, face-to-face interviews, laboratory tests and health examinations were performed at each clinical site of the study centers. A total of 3014 participants conducted a baseline examination. The mean age of participants was 76.0 ± 3.9 years (men: 47.5%). Among these participants recruited in 2016 and 2017, 2539 participants were followed up two years later (the follow-up rate in 2018 was 93.9%; in 2019 it was 96.2% and 37 participants died) [15]. Among them, participants with an absence of DEXA measurements, history of stroke or hemiplegia, diagnosed cognitive impairment or dementia (Mini-Mental Status Examination [MMSE] < 20) [16], incomplete physical function test, any fracture within one-year, hip or knee replacement, cancer treatment, or dependence needs for any activities of Korean instrumental activities of daily living (K-IADL) were excluded from this study [17]. Baseline participants who satisfied any one of the following criteria were also excluded: low appendicular skeletal muscle mass (ASM); low muscle strength; low physical performance according to the Asian Working Group for Sarcopenia (AWGS) 2019 diagnostic criteria (901 participants) (Figure 1) [18].

Finally, this study included 844 non-sarcopenic older adults (382 men; 462 women) who did not meet any diagnostic criteria for sarcopenia. The demographic data and medical history, including age, sex, education years, marital status, income per month, body mass index (BMI), smoking and alcohol status, and chronic diseases or comorbidities, were obtained from each participant. Smokers were defined as participants who smoked more than one cigarette in a week and alcohol consumers were defined as participants who drank alcohol more than once a week.

The KFACS protocol was approved by the Institutional Review Board (IRB) of the Clinical Research Ethics Committee of Kyung Hee University Medical Center (IRB number: 2015-12-103) and all participants provided written informed consent.

### 2.2. Vitamin B12

Participants’ blood samples were collected at the initial visit and B12 was measured by the Architect Vitamin Kit (Abbott Diagnostics, Lake Forest, IL, USA). The participants were divided by a serum B12 concentration cut-off of 350 pg/mL, which was reported to have a protective effect on myelin synthesis in the nervous system [7,19]. Individuals with a serum B12 concentration less than 350 (<350) pg/mL were defined as having B12 insufficiency. Participants were divided into clinically relevant categories by B12 concentration: insufficiency group (<350 pg/mL, equal to <258.3 pmol/L) and sufficiency group (≥350 pg/mL, ≥258.3 pmol/L).

### 2.3. Sarcopenia

Sarcopenia was diagnosed according to the AWGS 2019 criteria [20]. Participants with a low appendicular skeletal muscle mass (ASM) and either low muscle strength or poor physical performance were diagnosed with sarcopenia.

Muscle mass: We used DEXA to measure ASM and the appendicular skeletal muscle mass index (ASM/height^2^, ASMI) was calculated to compare the muscle mass at a different height. The cut-off values for low ASMI were <7.0 kg/m^2^ for men and <5.4 kg/m^2^ for women [21]. Of the eight centers, four used Hologic (Hologic Inc., Bedford, MA, USA) DXA systems and four used Lunar (GE Healthcare, Madison, WI, USA).

Muscle strength: Hand-grip strength (HGS) was measured by a hand dynamometer (Jamar, Bolingbrook, IL, USA). We measured HGS twice on both sides, with the elbow extended in a standing position. The participants held the grip for 3 s with full force and the maximum value was obtained (cut-off values: men, <28 kg; women: <18 kg).

Physical performance: We used a Short Physical Performance Battery (SPPB) to evaluate the physical performance. The SPPB is a well-accepted test for assessing lower extremity function in older adults. This test includes standing balance, 4-m usual gait speed and five counts of the sit-to-stand test. Each test was scored from 0 to 4 based on the reference values from the Established Populations for Epidemiologic Studies of the Elderly, with a maximum score of 12 points [22]. Participants who could not complete the sit-to-stand test were classified as failures. According to the AWGS 2019 diagnostic criteria, a score of ≤9 points was defined as low physical performance.

### 2.4. Statistical Analysis

Continuous variables were compared using Mann–Whitney U test or a t-test and categorical variables were compared by Pearson chi-squared test. Unadjusted and fully adjusted analyses were performed by logistic regression models and odds ratios (ORs) and 95% confidence intervals (CI) were calculated. Unadjusted and fully adjusted analyses were also calculated through generalized linear models and B estimates alongside their corresponding 95% CI values. The analyses were adjusted for potential confounding variables including age, dyslipidemia, hypertension, osteoporosis, osteoarthritis, diabetes mellitus, depression, BMI, smoking history, alcohol history, number of medications, MMSE-KC—Korean version and hemoglobin. The Statistical Package performed all statistical analyses for Social Sciences (version 25.0; SPSS Inc., Chicago, IL, USA) and *p* < 0.05 was defined as statistically significant.

## 3. Results

The baseline characteristics of the participants according to initial B12 levels are presented in Table 1. Among the 844 participants, 382 (45%) were men and 462 (55%) were women. In men, 366 (90%) were B12 sufficiency group and 40 (10%) were B12 insufficiency group. In women, 478 (92%) were B12 sufficiency group and 41 (8%) were B12 insufficiency group. The initial SPPB score was significantly lower in the B12 insufficiency group than that in the B12 sufficiency group in women (11.37 ± 0.77 vs. 11.10 ± 0.85, *p* < 0.05). Initial HGS and ASMI scores were not significantly different between the groups. Other characteristics such as age, BMI, alcohol use and diabetes mellitus were significantly higher in the B12 insufficiency group in women. The prevalence of osteoporosis was significantly higher in the B12 insufficiency group in men (Table 1).

Figure 2 shows the relative percentage change of sarcopenia parameters according to the initial B12 level over two years. All sarcopenia parameters decreased at follow-up. In both men and women, there were no significant differences in the percentage change (Δ) in the HGS or ASMI. On the other hand, SPPB showed a statistically significant reduction in the B12 insufficiency group in women only.

Table 2 shows the results of the generalized linear analysis for the relative percentage change of sarcopenia parameters according to the initial B12 level. The fully adjusted generalized linear model analysis showed that SPPB in women was significantly reduced in the B12 insufficiency group compared with that in the B12 sufficiency group (B estimate = −4.85, CI = −9.11 to −0.59). HGS and AMSI in women were more reduced in the B 12 insufficiency group compared with that in the B12 sufficiency group; however, it was not statistically significant.

Logistic regression analysis for the predictive power for sarcopenia of B12 insufficiency and its parameters according to sex is shown in Table 3. In women, the B12 insufficiency group had a significantly higher incidence of low SPPB (OR = 4.38, 95% CI = 2.15–8.84) and sarcopenia (OR = 5.90, 95% CI = 1.55–22.43) in the unadjusted and fully adjusted model, respectively. The B12 insufficiency group had a higher incidence of low HGS and ASMI. However, it was not statistically significant. 

## 4. Discussion

In this study, we investigated the longitudinal effects of B12 insufficiency on sarcopenia according to sex over two years. We found that B12 insufficiency negatively impacted physical performance measured by SPPB and increased the incidence of sarcopenia in women who were non-sarcopenic, even after adjusting for confounding factors. In contrast, B12 insufficiency had no apparent influence on the change of muscle mass, muscle strength, physical performance and incidence of sarcopenia in men.

Several studies have investigated the association between B12 levels and physical performance. However, this relationship still remains controversial. A cross-sectional study of 796 older adults investigated the association of homocysteine and B12 levels with gait and balance performance. Completed performance-oriented mobility assessments (POMA) of gait, balance and self-reports of instrumental activities of daily living (IADL) were done and the results showed that B12 level was not significantly related to POMA of balance, POMA of gait and IADL scores [23]. In addition, Vidoni et al. conducted a longitudinal analysis to assess the association of B12 serum levels with gait speed decline. B12 serum levels were not significantly associated with a decline in gait speed over an average of 5.4 years [24]. However, a few studies have found that B12 levels are related to physical performance. In a cross-sectional study of 703 community-dwelling Caucasian older women, Matteini et al. revealed that low B12 levels contributed to frailty syndrome defined by low HGS, endurance, physical activity and walking speed [25]. In addition, Oberlin BS et al. assessed whether self-reported disability (including ADL), balance and gait speed are associated with low B12 levels and reported that low B12 levels are associated with the disability in ADL and reduced mobility [26]. In this study, we found that B12 insufficiency was inversely related to physical performance as measured using the SPPB. The discrepancy between the results of a few previous studies and our study may be due to the difference in the criteria for defining B12 deficiency and the method of measuring physical performance. The definition of B12 deficiency in previous studies varied from 200 to 350 (pg/mL). While previous studies investigated the relationship between B12 deficiency and physical performance by balance, gait speed, or ADL scores, we used the SPPB, which is a group of measures combining the results of gait speed, balance and repeated chair stands. The SPPB is more complex, but can accurately and comprehensively measure physical function.

A decline in physical performance may be due to neurologic complications related to B12 insufficiency. B12 insufficiency can cause myelin damage due to deficient methylation of myelin protein. It has been reported that a lack of B12 can induce several neurologic complications, including myelopathy and peripheral neuropathy [27,28]. Subacute combined degeneration, characterized by demyelination of the posterior and lateral columns of the spinal cord, is often found in patients with low B12 levels. It commonly presents with impairment of position sense, paresthesia, ataxia and gait disturbance [29,30]. Peripheral neuropathy has also been reported as another neurological complication caused by a lack of B12. According to previous electrodiagnostic studies, patients with B12 deficiency-related neuropathy typically have a sensorimotor axonal neuron defect with some demyelinating features. In a case series of nine patients with B12 deficiency-related neuropathy, four had sensorimotor (predominantly sensory) axonal polyneuropathy and five had sensory neuronopathy [31,32]. In a previous study, peripheral neuropathy in older adults was associated with a deterioration in physical performance [33]. The effects of neurologic complications, especially in sensory nerves caused by B12 insufficiency, may have resulted in a decline in SPPB.

Although it was not statistically significant, ASMI and HGS decreased more in the B12 insufficiency group than in the B12 sufficiency group in women over two years. Previous studies have reported that low B12 levels negatively affect ASMI and HGS. Gedmantaite et al. investigated the association between diet and HGS and showed a positive correlation between B12 intake and HGS in women [34]. In a prospective study of 403 older adults aged >60 years, total skeletal mass, lean body mass and skeletal muscle mass index were lower in the B12 insufficient group than in the B12 sufficient group [8]. Furthermore, in a previous cross-sectional study of 2325 community dwellers in Korea, B12 insufficiency was associated with a high prevalence of low ASMI, but not SPPB [14]. The following explanations can explain the discrepancies in results: As a limitation of the previous cross-sectional study, it was difficult to clarify the longitudinal relationship between B12 insufficiency and sarcopenia over time. Since a short period of two years may not be enough for B12 insufficiency group to affect ASMI and HGS, a study of longer duration may show statistically significant results. Moreover, a previous study included all community-dwelling individuals. However, this study excluded sarcopenic participants with a risk of sarcopenia who fulfilled any one of the criteria of sarcopenia. Therefore, there was a difference in the baseline population, which may have caused discrepancies. Furthermore, in this study, analysis was performed according to sex, whereas in previous studies, analysis was not performed by sex. Because sarcopenia is defined as low ASMI and either low HGS or physical performance, the effect of B12 insufficiency on these parameters may increase the incidence of sarcopenia in women. Accordingly, the impact of B12 insufficiency seems to act on both muscle mass and physical performance but varies depending on the sarcopenic health status and sex.

In this study, B12 insufficiency negatively impacted physical performance and the incidence of sarcopenia in women only. This indicates that the lack of B12 in women significantly affects physical performance at both baseline and the degree of decline over time. Although the reason for these conflicting results between the sexes is not known, the following gonadal hormone hypotheses could partially explain. Testosterone, the male sex steroid hormone, is much higher in men than in women and it is known to play an important role in central nervous system development [35,36,37]. One of the lesser known actions of testosterone is neuroprotection by mediating neuronal differentiation and increasing neurite overgrowth via activation of androgen pathways [38]. Furthermore, previous studies have revealed that testosterone has protective effects against spinal cord injury induced by glutamate and ischemia/reperfusion and reduces the extent of spinal cord damage [39]. This suggests that axonal injury may occur only in low B12 settings as well as low testosterone levels, such as in women, suggesting that B12-induced axonal injury may be less pronounced in men with relatively high testosterone levels. Therefore, in men, B12 insufficiency alone may not be associated with sarcopenia because of the neuroprotective effects of testosterone. However, additional in vivo or randomized controlled studies are needed to support our hypothesis.

This study had several limitations. First, the intake of B12 supplements was not considered in this study. Because B12 is included in commercial multivitamin supplements, it may be helpful to investigate the effects of the intake of these supplements. Second, B12 insufficiency was defined only by serum B12 levels without considering other B12 insufficiency markers, including methylmalonic acid and homocysteine, which reflect the biochemical action of B12. Third, the amount of protein intake, nutrition status and daily physical activity of the participants were not investigated. Fourth, the number of participants in the B12 insufficiency group was relatively small (40 men and 41 women) compared to the B12 sufficiency group. Fifth, peripheral neuropathy induced by B12 insufficiency can impair physical performance. However, since we did not perform an electrical diagnostic study, such as a nerve conduction study, we could not confirm peripheral neuropathy. Finally, the two-year follow-up period was relatively short. A longitudinal study with a longer duration may demonstrate a significant effect of B12 insufficiency on other component parameters of sarcopenia.

## 5. Conclusions

In conclusion, this study was the first longitudinal cohort study to investigate the association between B12 insufficiency and component parameters of sarcopenia in non-sarcopenic older adults. In this study, we found that B12 insufficiency negatively impacts physical performance defined as SPPB and increases the incidence of sarcopenia based on the AWGS 2019 criteria only in women. On the contrary, B12 insufficiency had no apparent influence on the incidence of sarcopenia in men.

## Figures and Tables

**Figure 1 nutrients-15-00936-f001:**
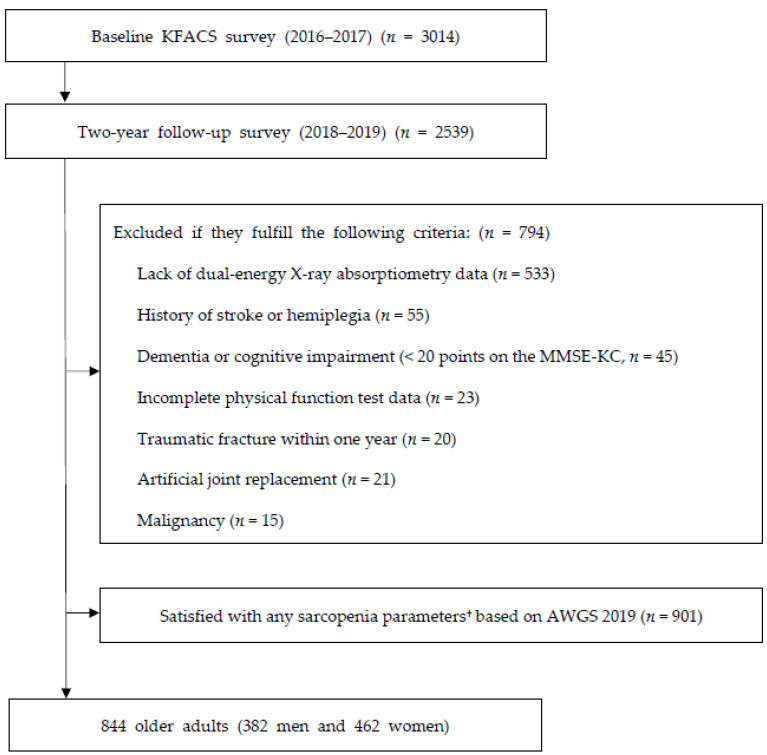
Flow chart of the participant recruitment process. ^†^ Sarcopenia parameters: Low Appendicular skeletal muscle mass index (ASMI) (<7.0 kg/m^2^ for men and <5.4 kg/m^2^ for women), low hand-grip strength (HGS) (<28 kg for men and <18 kg for women) and low short physical performance battery score (SPPB) (≤9 for both sexes). Abbreviations: KFACS, Korean Frailty and Aging Cohort Study; MMSE-KC, Mini-Mental Status Examination in the Korean version of the CERAD assessment packet; ADL, activities of daily living; AWGS, Asian Working Group for Sarcopenia.

**Figure 2 nutrients-15-00936-f002:**
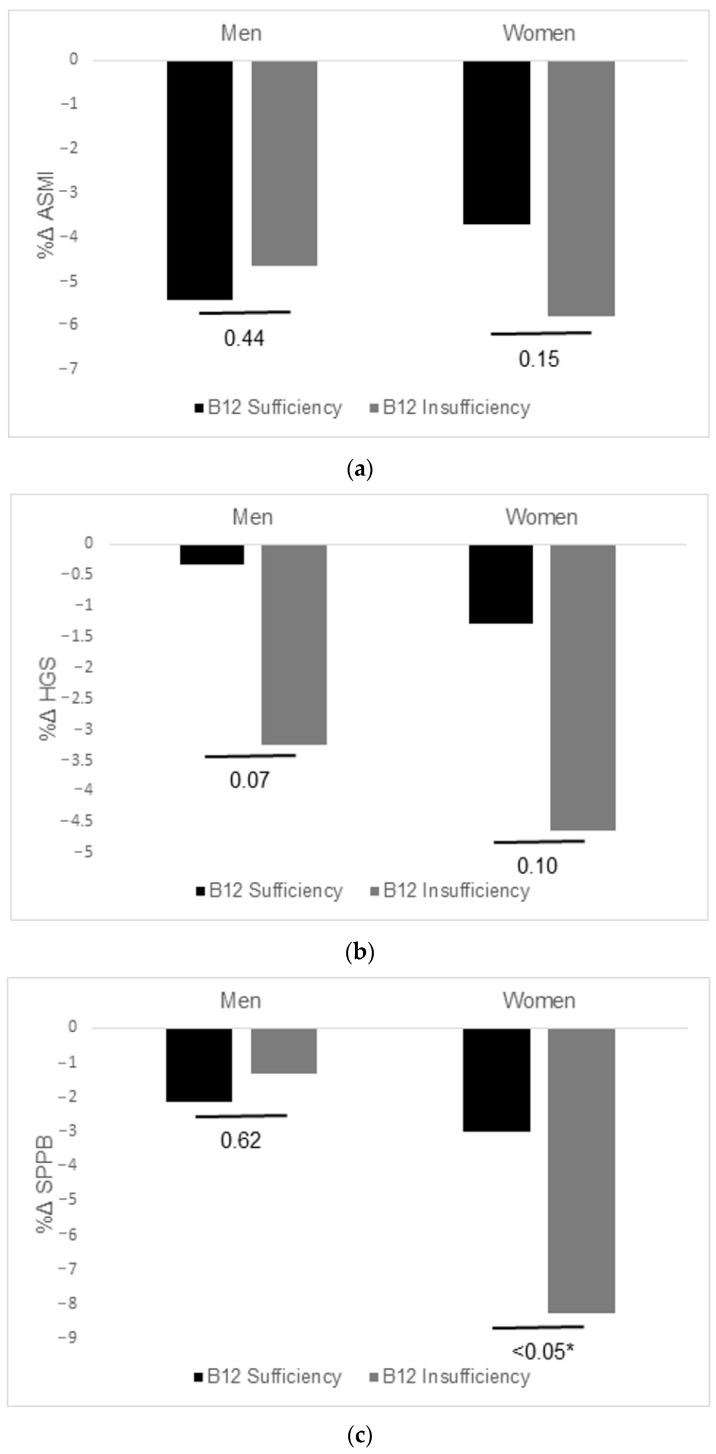
Changes in sarcopenia parameters according to vitamin B12 level by sex. (**a**) Relative percentage change in appendicular skeletal muscle mass index (ASMI). (**b**) Relative percentage change in hand-grip strength (HGS). (**c**) Relative percentage change in short physical performance battery score (SPPB). * *p* < 0.05.

**Table 1 nutrients-15-00936-t001:** Baseline characteristics of participants by vitamin B12 level according to sex.

Characteristics	Men			Women		
	B12 Sufficiency † (*n* = 366)	B12 Insufficiency † (*n* = 40)	*p*	B12 Sufficiency † (*n* = 478)	B12 Insufficiency † (*n* = 41)	*p*
Age (years)	75.28 ± 3.56	75.49 ± 3.73	0.72	74.69 ± 3.53	76.46 ± 3.60	<0.01 *
BMI (kg/m^2^)	25.15 ± 2.40	25.90 ± 2.62	0.06	25.26 ± 2.70	26.22 ± 3.34	<0.05 *
Education years (*n*, %)						
≤6	91 (26.7)	11 (26.8)	0.82	240 (56.7)	16 (41.0)	0.15
7–12	152 (44.6)	20 (48.8)		150 (35.5)	18 (46.2)	
>13	98 (28.7)	10 (24.4)		33 (7.8)	5 (12.8)	
Marriage (*n*, %)						
Married	314 (92.1)	38 (92.7)	0.89	213 (50.4)	20 (51.3)	0.91
Not married	27 (7.9)	3 (7.3)		210 (49.6)	19 (48.7)	
Income per month (Korean million won) (*n*, %)						
>3	93 (27.3)	12 (29.3)	0.23	54 (12.8)	4 (10.3)	0.90
1–3	144 (42.2)	12 (29.3)		126 (36.9)	15 (38.5)	
<1	104 (30.5)	17 (41.5)		213 (50.4)	20 (51.3)	
Current smoker (*n*, %)	31 (9.1)	5 (12.2)	0.52	1 (0.2)	0 (0.0)	0.76
Alcohol use (*n*, %)	241 (70.7)	28 (68.3)	0.75	268 (63.4)	32 (82.1)	<0.05 *
Hypertension (*n*, %)	172 (50.4)	25 (61.0)	0.20	256 (60.5)	27 (69.2)	0.29
Dyslipidemia (*n*, %)	93 (27.3)	13 (31.7)	0.55	168 (39.7)	18 (46.2)	0.43
Diabetes mellitus (*n*, %)	70 (20.5)	13 (31.7)	0.10	69 (16.2)	12 (30.8)	<0.05 *
Depression (*n*, %)	4 (1.2)	1 (2.4)	0.50	3 (0.7)	0 (0.0)	0.60
OA (*n*, %)	36 (10.6)	8 (19.5)	0.09	136 (32.2)	10 (25.6)	0.40
Osteoporosis (*n*, %)	5 (1.5)	4 (9.8)	<0.01 *	89 (21.0)	10 (25.6)	0.50
Hb (g/dL)	14.33 ± 1.33	14.15 ± 1.09	0.41	12.93 ± 1.02	12.68 ± 1.37	0.27
MMSE-KC	26.86 ± 2.18	26.71 ± 2.41	0.68	26.10 ± 2.53	26.54 ± 2.29	0.30
HGS (kg)	35.58 ± 4.63	35.03 ± 5.71	0.48	22.96 ± 2.95	22.42 ± 3.11	0.28
ASMI (kg/m^2^)	7.71 ± 0.55	7.69 ± 0.69	0.84	6.16 ± 0.59	6.25 ± 0.63	0.35
SPPB	11.56 ± 0.69	11.61 ± 0.70	0.66	11.37 ± 0.77	11.10 ± 0.85	<0.05 *

Abbreviations: B12, vitamin B12; BMI, body mass index; OA, osteoarthritis; Hb, hemoglobin; MMSE-KC, Mini-Mental Status Examination in the Korean version of the CERAD assessment packet; HGS, hand-grip strength; ASMI, appendicular skeletal muscle mass index; SPPB, short physical performance battery. † B12 sufficiency, B12 concentration ≥ 350 pg/mL; B12 insufficiency, B12 concentration < 350 pg/mL. * *p* < 0.05.

**Table 2 nutrients-15-00936-t002:** Generalized linear models for change of each parameter according to sex.

Variables	B12 Sufficiency (≥350 pg/mL)	B12 Insufficiency (<350 pg/mL)			
		Men		Women	
		Unadjusted	Fully Adjusted	Unadjusted	Fully Adjusted
		B Estimate (CI)	B Estimate (CI)	B Estimate (CI)	B Estimate (CI)
Change of HGS (%)	Reference	−2.92 (−6.1 to 0.26)	−2.61 (−5.82 to 0.61)	−3.35 (−7.33 to 0.64)	−3.12 (−7.22 to 0.98)
Change of ASMI (%)	Reference	0.97 (−1.51 to 3.45)	1.03 (−1.42 to 3.48)	−2.09 (−4.95 to 0.77)	−1.89 (−4.81 to 1.03)
Change of SPPB (%)	Reference	0.86 (−2.53 to 4.25)	1.47 (−1.93 to 4.86)	−5.27 (−9.48 to −1.63) *	−4.85 (−9.11 to −0.59) *

Abbreviations: CI, confidence interval; HGS, hand-grip strength; ASMI, appendicular skeletal muscle mass index; SPPB, short physical performance battery. The fully adjusted model was adjusted for age, body mass index, hypertension, dyslipidemia, osteoarthritis, osteoporosis, diabetes mellitus, depression, smoking history, alcohol history, number of medications, MMSE-KC score and hemoglobin. * *p* < 0.05.

**Table 3 nutrients-15-00936-t003:** Logistic regression analysis of vitamin B12 insufficiency (<350 pg/mL) predicting sarcopenia and its parameters according to sex.

Variables	Unadjusted Model		Fully Adjusted Model	
	Men	Women	Men	Women
	OR (95% CI)	OR (95% CI)	OR (95% CI)	OR (95% CI)
Muscle strength				
Low HGS †	2.26 (0.87–5.92)	1.84 (0.73–4.68)	1.87 (0.64–5.43)	1.72 (0.63–4.70)
Muscle mass				
Low ASMI †	0.58 (0.27–1.26)	1.23 (0.56–2.69)	0.51 (0.22–1.22)	1.67 (0.56–3.34)
Physical performance				
Low SPPB †	0.50 (0.11–2.15)	4.36 (2.15–8.84) **	0.38 (0.07–1.94)	4.38 (2.01–9.55) **
Sarcopenia ††	1.27 (0.36–4.46)	2.96 (1.05–8.39) *	0.81 (0.18–3.73)	5.90 (1.55–22.43) *

Abbreviations: OR, odds ratio; CI, confidence interval; HGS, hand-grip strength; ASMI, appendicular skeletal muscle mass index; SPPB, short physical performance battery. The fully adjusted model was adjusted for age, body mass index, hypertension, dyslipidemia, osteoarthritis, osteoporosis, diabetes mellitus, depression, smoking history, alcohol history, number of medications, MMSE-KC score and hemoglobin. † Low HGS, <28 kg for men and <18 kg for women; low ASMI, <7.0 kg/m^2^ for men and <5.4 kg/m^2^ for women; low SPPB, score ≤ 9 for both sexes. †† Sarcopenia: low ASMI (<7.0 kg/m^2^ for men and <5.4 kg/m^2^ for women) and either a low HGS (<28 kg for men and <18 kg for women) or low physical performance (SPPB score ≤ 9 for both sexes). * *p* < 0.05. ** *p* < 0.01.

## Data Availability

All cohort data supporting this study’s findings are available from the KFACS and are open to all researchers on reasonable request. All published articles and news articles using the KFACS database, data provision manuals and contact information are available on the KFACS website (http://www.kfacs.kr (accessed on 3 July 2022).

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
