# Peer review of "Impact of Vitamin B12 Insufficiency on the Incidence of Sarcopenia in Korean Community-Dwelling Older Adults: A Two-Year Longitudinal Study"

_nutrients, 2023, doi:10.3390/nu15040936_

Round 1
Reviewer 1 Report
nutrients-2198084-peer-review-v1
Impact of Vitamin B12 Insufficiency on the Incidence of Sarcopenia in Korean Community-Dwelling Older Adults: A Two-Year Longitudinal Study
The study investigated the impact of B12 levels on alterations in muscle mass, function and strength over two years. Non-sarcopenic older adults (n = 926) aged 70–84 were included. In women, multivariate analysis showed that the B12 insufficiency group had a significantly higher incidence of low SPPB scores and sarcopenia. No such association was observed in men.
It seems that this is the first longitudinal cohort study investigating the association between B12 insufficiency and component parameters of sarcopenia in non-sarcopenic older adults.
The manuscript is generally well-written and data adequately presented and discussed.
The fact that B12 deficiency groups were rather limited (40 and 41 subjects), should be mentioned in the limitations of the study. Likewise, non-consistent changes in HGS vs ASMI and SPBB in men.
Author Response
Reviewer 1
The study investigated the impact of B12 levels on alterations in muscle mass, function and strength over two years. Non-sarcopenic older adults (n = 926) aged 70–84 were included. In women, multivariate analysis showed that the B12 insufficiency group had a significantly higher incidence of low SPPB scores and sarcopenia. No such association was observed in men.
It seems that this is the first longitudinal cohort study investigating the association between B12 insufficiency and component parameters of sarcopenia in non-sarcopenic older adults.
The manuscript is generally well-written and data adequately presented and discussed.
The fact that B12 deficiency groups were rather limited (40 and 41 subjects), should be mentioned in the limitations of the study. Likewise, non-consistent changes in HGS vs ASMI and SPBB in men.
Response: Thank you for your thoughtful suggestion. We agree with you and have added this content in line numbers 270-272.
Fourth, the number of participants in the B12 insufficiency group was relatively small (40 men and 41 women) compared to the B12 sufficiency group.
Likewise, non-consistent changes in HGS vs ASMI and SPBB in men.
Response: Thank you for your thoughtful suggestion. Relative percentage change of sarcopenia defining parameters (HGS, ASMI, and SPBB) according to the initial B12 level over two years are explained in figure 2 as follows.
Figure 2 shows the relative percentage change in sarcopenia parameters according to the initial vitamin B12 level over two years. All sarcopenia parameters decreased at follow-up. In both men and women, there were no significant differences in the percentage change in HGS or ASMI between the sufficient and insufficient groups. On the other hand, SPPB showed a statistically significant reduction in the B12 insufficiency group in women only.

Reviewer 2 Report
Comments to the Author
This is a practically important original article indicates that “Impact of Vitamin B12 Insufficiency on the Incidence of Sarcopenia in Korean Community-Dwelling Older Adults: A Two-Year Longitudinal Study”.
However, it is necessary to reexamine the research method etc. in several respects.
1. Title
This is a practically important original article indicates that “Impact of Vitamin B12 Insufficiency on the low physical performance in Korean Community-Dwelling Older Adults: A Two-Year Longitudinal Study”.
2. Materials and Methods, Data and study population
Were participants in this study included who had malabsorption of vitamin B12, such as autoimmune metaplastic atrophic gastritis or inflammatory bowel disease?
3. Materials and Methods, Sarcopenia
Did this study measure body composition and skeletal muscle mass during fasting?
4. Result
There were 40 males and 41 females with vitamin B12 deficiency, respectively. Were participants with vitamin B12 deficiency such as peripheral neuropathy or megaloblastic anemia included in this study? 
Discussion explains that vitamin B12 deficiency and diseases such as peripheral neuropathy are associated with decreased physical function.
5. Discussion
The discussion was unclear because the results of the present study were not consistent with the results of previous studies that reported vitamin B12 and physical performance.
In addition, the authors considered the following.
"The discrepancy in these results may be due to the difference in the criteria of B12 deficiency and the measurement method of physical performance". This discussion is unclear.
6. Discussion
In this study, there were sex differences in the association between vitamin B12 deficiency and physical performance. Testosterone levels were not assessed in this study, and the mechanisms between sex steroid hormone and neurological function, vitamin B12, and physical performance are unclear. In conclusion, sex differences are unknown.
7. Discussion, limitation
It should be indicated that physical activity was not investigated in the limitation of discussion.na
Author Response
Reviewer 2
This is a practically important original article indicates that “Impact of Vitamin B12 Insufficiency on the Incidence of Sarcopenia in Korean Community-Dwelling Older Adults: A Two-Year Longitudinal Study”.
However, it is necessary to reexamine the research method etc. in several respects.
- Title
This is a practically important original article indicates that “Impact of Vitamin B12 Insufficiency on the low physical performance in Korean Community-Dwelling Older Adults: A Two-Year Longitudinal Study”.
- Materials and Methods, Data and study population
Were participants in this study included who had malabsorption of vitamin B12, such as autoimmune metaplastic atrophic gastritis or inflammatory bowel disease?
Response: Thank you for your comment. Unfortunately, specific information about chronic diseases causing vitamin B12 (B12) malabsorption was not included in the data from the Korean Frailty and Aging Cohort Study (KFACS). Therefore, it is unclear whether participants with B12 malabsorption were included in the study. However, since the objective of this study is to investigate the association of B12 levels and sarcopenia, we thought that the presence or absence of B12 malabsorption in the participants was not significant. Thank you.
- Materials and Methods, Sarcopenia
Did this study measure body composition and skeletal muscle mass during fasting?
Response: Thank you for your comment. When measuring body composition by bioelectrical impedance analysis (BIA), fasting is essential for accuracy. Since we used dual-energy X-ray absorptiometry (DEXA) to measure body composition, fasting status was not considered essential before body composition analysis in this study. Thank you.
- Result
There were 40 males and 41 females with vitamin B12 deficiency, respectively. Were participants with vitamin B12 deficiency such as peripheral neuropathy or megaloblastic anemia included in this study? 
Response: Thank you for your thoughtful comment. Among the participants included in this study, there might be some participants with macrocytic anemia. We thought that megaloblastic anemia induced by B12 deficiency could negatively impact physical performance, so the hemoglobin level was adjusted in logistic regression analysis to address confounding bias. Thank you
Discussion explains that vitamin B12 deficiency and diseases such as peripheral neuropathy are associated with decreased physical function.
Response: Thank you for your thoughtful advice. Unfortunately, tests such as electrophysiologic study to diagnose peripheral neuropathy were not conducted in participants of the KFACAS. Therefore, it is unknown whether participants with peripheral neuropathy were included in this study. We found that B12 insufficiency negatively impacts physical performance and increases the incidence of sarcopenia in this study, so we have tried to explain our results based on previous studies of B12 deficiency neuropathy. We have added following contents in limitation on line number 272-275.
Fifth, peripheral neuropathy induced by vitamin B12 insufficiency can impair physical performance. However, since we did not perform an electrical diagnostic study, such as a nerve conduction study, we could not confirm peripheral neuropathy.
- Discussion
The discussion was unclear because the results of the present study were not consistent with the results of previous studies that reported vitamin B12 and physical performance.
In addition, the authors considered the following.
"The discrepancy in these results may be due to the difference in the criteria of B12 deficiency and the measurement method of physical performance". This discussion is unclear.
Response: Thank you for your thoughtful comment. We agree with you and have added this content in line number 198-206 to clearly describe the reasons for discrepancy between a few previous studies and our study.
The discrepancy between the results of a few previous studies and our study may be due to the difference in the criteria for defining B12 deficiency and the method of measuring physical performance. The definition of B12 deficiency in previous studies varied from 200 to 350 (pg/mL). While previous studies investigated the relationship between B12 deficiency and physical performance by balance, gait speed, or ADL scores, we used the SPPB, which is a group of measures combining the results of gait speed, balance, and repeated chair stands. The SPPB is more complex, but can accurately and comprehensively measure physical function.
- Discussion
In this study, there were sex differences in the association between vitamin B12 deficiency and physical performance. Testosterone levels were not assessed in this study, and the mechanisms between sex steroid hormone and neurological function, vitamin B12, and physical performance are unclear. In conclusion, sex differences are unknown.
Response: Thank you for your thoughtful comment. We agree with you our hypothesis that testosterone may have a neuroprotective effect and protective impact on sarcopenia in men is unclear.
We thought that it may be necessary to explain the sex differences in the results, so we have modified the paragraph to say that the reason for conflicting results between the sexes is not yet known, and the hypothesis we mentioned could partially explain the results. (Line number 250-254)
Because we did not assess testosterone level, we have added reference articles that investigated the testosterone level by sex to support our hypothesis.
Although the reason for these conflicting results between the sexes is not known, the following gonadal hormone hypotheses could partially explain it. Testosterone, the male sex steroid hormone, is much higher in men than in women, and it is known to play an important role in central nervous system development 35-37.
- Discussion, limitation
It should be indicated that physical activity was not investigated in the limitation of discussion.na
Response: Thank you for your comment. We agree with you and have added the following sentence in line numbers 269-270.
Third, the amount of protein intake, nutrition status, and daily physical activity of the participants were not investigated.

Round 2
Reviewer 2 Report
No comment